# Energy Efficient Data Dissemination for Large-Scale Smart Farming Using Reinforcement Learning

Muhammad Yasir Ali [1], Abdullah Alsaeedi [2],*, Syed Atif Ali Shah [1,3], Wael M. S. Yafooz [2],*
and Asad Waqar Malik [4]

1 Department of Computer Science & Artificial Intelligence, Air University, Islamabad 44230, Pakistan
2 Department of Computer Science, College of Computer Science and Engineering, Taibah University, Medina 42353, Saudi Arabia
3 Department of Computer Science, Al-Madinah International University, Kuala Lumpur 57100, Malaysia
4 Department of Computer Science, National University of Sciences and Technology (NUST) Islamabad Pakistan, Islamabad 44000, Pakistan
* Correspondence: aasaeedi@taibahu.edu.sa (A.A.); wyafooz@taibahu.edu.sa (W.M.S.Y.)

**Abstract:** Smart farming is essential to increasing crop production, and there is a need to consider the technological advancements of this era; modern technology has helped us to gain more accuracy in fertilizing, watering, and adding pesticides to the crops, as well as monitoring the conditions of the environment. Nowadays, more and more sophisticated sensors are being developed, but on a larger scale, agricultural networks and the efficient management of them is very crucial in order to obtain proper benefits from technology. Our idea is to achieve sustainability in large-scale farms by improving communication between wireless sensor nodes and base stations. We want to increase communication efficiency by introducing machine learning algorithms. Reinforcement learning is the area of machine learning which is concerned with how involved agents are supposed to take action in specified environments to maximize reward and achieve a common goal. In our network, a large number of sensors are being deployed on large-scale fields; reinforcement learning is used to find the optimal set of paths towards the base station. After a number of successful paths have been developed, they are then used to transmit the sensed data from the fields. The simulation results have shown that in larger scales, our proposed model had less transmission delay than the shortest path transmission model and broadcasting techniques that were tested against the data transmission paths developed by reinforcement learning.

**Keywords:** wireless sensor network; reinforcement learning; simulation; AnyLogic; Agent Based Modelling

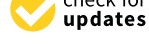



## 1. Introduction

The exponential increase in the human population has had a direct impact on food, resulting in increased demand for agricultural resources. It has been estimated that by 2050, the world's population can hit around 9.7 billion [1]. However, due to food scarcity, more than 10 million people will die from hunger every year [2]. Therefore, to handle this crisis situation, improving the crop production from already available land is necessary to fulfill the growing need [1]. Technological advancements in IoT have already helped farmers to manage their crop production effectively [1]. Many farmers have adopted the IoT-based network for crop management [3]. In the existing system, wireless sensors are used to gather real-time crop data; however, due to limited capacity and energy constraints, the system faces several limitations. Further, recharging sensors or changing the batteries of the devices deployed in vast and humid fields is an unfeasible solution. Moreover, it also requires technical expertise to operate. Therefore, there is a need for the efficient utilization of battery-operated sensors/devices for the sustainable network. The energy consumption of a sensor is directly proportional to its active mode, data processing, transmission, and

reception [4]. Notably, the sensors are also used for ad-hoc communication, and act as a relay node which also impacts their energy [3]. The efficient utilization of sensors for large-scale farming has not been explored in detail. The work mostly focuses on sensor deployment without considering the data transmission strategy; this makes it unsuitable for large-scale farming because of cost and administrative reasons. In large-scale farming, the covering of the entire area with a large number of interconnected devices is required [5]. Thus, deploying sensors with efficient communication enhances network lifetime.

Recent developments in wireless sensor networks and their applications have created a big demand for bandwidth and power resources [6]. In order to collect a large amount of data from the fields, the deployment of a large number of sensor nodes spanning a large area is required. However, deploying a massive number of sensors communicating over an ad-hoc wireless sensor network triggers channel access, organization, and interference problems [6]. Moreover, in precision agriculture, increasing agricultural productivity techniques is proposed to improve the precision farming field, due to which additional delays occur in providing farmers with the correct information. Additional energy consumption is created as well [7]. From one point of view, wireless sensor networks are a technological advancement, but on the other hand, the acceptance rate of wireless sensor technologies is strongly affected by the perceptions of farmers who are the significant adopters of the latest technologies in the wireless network domain. Yet, the way WSN technology is perceived has been poorly studied [8]. The authors argue on the reasons for which presently, WSNs are not integrated in precision agriculture on a larger scale [9]. In the existing work, smart farming has been considered either on small-scale fields where sensors are deployed in a single farm [10]. Considering a single farm reduces the data dissemination issues that require additional energy for relay purposes. Thus, applying the same approach to large-scale connected farms raises serious energy issues. Moreover, the other contribution on a large-scale focused on data gathering to perform analytics or predictions without considering the underlying network [11]. Thus, for every IoT-based system deployment, a comprehensive networking model is required to support the best possible deployment scenario for large-scale farming. There are numerous ways for data to be transmitted in a wireless network; however, machine learning/AI-based algorithms complement the system by predicting and adopting the best possible method based on the available resources.

The IoT-based network is data-driven, and more context and situation-aware [3]. The sensor data helps in crop monitoring and is also used for dynamic network management. The paper is concerned with improving wireless sensor communication in large farms that have utilized IoT technology to monitor the crop conditions. This is performed with the help of machine learning algorithms, more specifically reinforcement learning, applied on a large network of sensors, with the purpose being to improve communication with the control station. The proposed solution manages to find successful path to the base station with a lesser delay than the shortest path transmission model. We propose finding the optimal set of paths using a reinforcement learning algorithm. The aim is to achieve sustainability for the farming network. The main features of the proposed system are listed below: The model provides network connectivity for static sensor nodes deployed in large-scale fields. Multi-hop communication is provided by sending data towards edge nodes (deployed on the edge of each field), which forward data to the base station. We add a reinforcement learning [12] algorithm at the edge node that finds a set of suitable paths with the geographic reference to each node. These paths are used to transmit data.

This paper is organized as follows. Section 2 covers the existing smart farming methods. Section 3 covers the system model and proposed technique. The experimental details and evaluation are covered in Section 4. Finally, Section 5 have conclusions and the future prospects for the work conducted, respectively.

## 2. Related Work

Some of the literature reviewed focus on low-power wireless sensor networks, due to the fact that data transmission is often energy-consuming, and sensors rely on batteries

or researchable power which is limited. This is performed by using specific node-based transmission or sampling rates, with different algorithmic approaches. Other studies focus on network configuration or lifetime optimization to avoid network congestion.

Srbinovska et al. [13] proposed a wireless sensor network, which is deployed to capture the information about environmental parameters that affect the crops. The objective was to develop a low-power wireless sensor network that can live a longer life to collect information about the crops. They investigated the wireless sensor network and power consumption of nodes in detail to achieve longer network life and lower power consumption among the nodes that reside in the network. Their aim was to develop a low-cost non-robust wireless sensor network that can be used for data collection of crops in greenhouses. They collected data from their deployed network, conducted a theoretical analysis of the power consumption from their network, and it compared with quantitative analysis from their deployed wireless sensor networks. According to Srbinovska et al., the sensors should keep it in sleeping mode as long as some external event triggers them to keep it in the processing mode where the sensors are about to sense, collect, and compute the data from the environment they are deployed in. The power consumption results from their deployed network were similar to the estimated theoretical analysis that they made.

Jain et al. [14] proposed an energy-efficient data collection from the wireless sensor network that involves a gateway in an agricultural environment. The sensor network optimizes itself to collect data from the nodes, and in this way is energy efficient. The gateway of the wireless sensor network adapts itself for the parameters that need to be sensed, and the data is collected by the sensor nodes. The adaptive learning of the gateway depends on the spatial and temporal crop characteristics that may change during the complete season. The gateway was introduced with supervised learning principles that were meant to set specific parameters for the sensor nodes. According to Jain et al. energy efficiency can be improved by specifically setting the sensing parameters at the gateway beforehand, and having them communicate with the sensor nodes so that they can sense and collect data about the very specific parameters needed. They analyzed the model and compared it with non-adaptive models as well. In non-adaptive models, the data transmission occurred at a continuous rate.

The adaptive model had successfully shown the classification of data points for diseased and non-diseased crops whose parameters were set by supervised learning at the gateway. With different variations in the data set, the model was able to dynamically set the parameters for the crop. Results showed that the proposed adaptive sampling model based on random forest can significantly reduce energy consumption by varying the sampling rate of the sensor nodes. The adaptive model which was set on the basis of the random forest algorithm gave 22 percent to 30 percent energy efficiency for the sampling rates kept more frequent than other techniques. Future work includes deployment of the adaptive model.

Sathiya et al. [10] designed a wireless sensor network with nodes and a base station using zigbee. The wireless sensor network had seven nodes comprised of six sensor nodes and one coordinator. The wireless sensor network was aimed at small-scale fields to sense and collect data about the crop. They successfully implemented the model collector results from the model. As their future work suggests, the nodes should be placed properly to achieve efficiency. Power consumption is also suggested as an extended work.

Zhang et al. [11] developed a wireless underground sensor network. They had a fully operational network and were able to collect data on the website server. The network had a back-end system, a base station, and a set of sensor nodes. Each sensor node had two sensor components which were meant for collecting data for four crop properties: temperature, soil moisture, volumetric water content, and electric conductivity. As a result, they collected a great deal of information, and analyzed it at the back end. The network was meant for small-scale fields, and in future work, they left it for larger-scale work and networking; energy efficiency parameters were also to be considered.

Similarly, most of the previously-preformed experiments and simulations were conducted either on a small scale or, if undertaken on a large scale, there was no network optimization for network performance. The summary of related work is given in the Table 1, where the comparison is conducted based on the required parameters of network performance optimization.

This study aims to propose a sustainable wireless sensor network from all perspectives (energy, communication, configuration), by training agents through reinforcement learning to find an optimal path based on specific rewards (such as the shortest path from sensor node to edge node), taking in consideration the energy spent when transmitting data (also 'rewarding' agents with units). Reinforcement learning is an unsupervised type of machine learning [15] where the agents learn by trial and error. Generally, machine learning algorithms work well in congested networks by locating high-quality links [16]. Mainly reinforcement learning has been applied to make big data analysis from the crop data. Algorithms have also been applied to sensors to gather data to make optimal resource usage. Reinforcement learning has also proven successful in sampling data for crop fields from fully autonomous aerial scouting methods [17]. Reinforcement learning has been applied to agricultural wireless sensor networks for various reasons. Routing is one of the biggest problems, and a solution is given using a reinforcement learning meta-routing strategy [18]. Similarly, reinforcement learning has been applied to make smart decisions for irrigation [2], and also to determine crop production for a season.

**Table 1.** Summary of work related to wireless sensor network incorporated in smart agriculture.

| Authors | Mobility | Scale | Hop Count | Energy Efficiency | Network Lifetime Optimization | Network Configuration |
|---|---|---|---|---|---|---|
| Sribnovska et al. [13] | ✗ | Small | Single | ✔ | ✗ | ✗ |
| Sathiya et al. [10] | ✗ | Small | Multi | ✗ | ✗ | ✗ |
| Zhang et al. [11] | ✗ | Large | Single | ✗ | ✗ | ✗ |
| Khan et al. [19] | ✗ | Small | Multi | ✗ | ✗ | ✗ |
| Granda et al. [20] | ✗ | Small | Multi | ✗ | ✗ | ✗ |
| Patokar et al. [21] | ✗ | Small | Single | ✗ | ✗ | ✗ |
| Jain et al. [14] | ✗ | Small | Single | ✔ | ✔ | ✔ |
| Nurellari et al. [22] | ✗ | Small | Single | ✔ | ✔ | ✗ |
| Hu et al. [23] | ✗ | Small | Multi | ✗ | ✔ | ✔ |
| Rathinam et al. [24] | ✗ | Small | Multi | ✗ | ✗ | ✗ |
| Sharma et al. [25] | ✗ | Small | Multi | ✗ | ✔ | ✔ |
| Ahmed et al. [26] | ✗ | Small | Multi | ✗ | ✗ | ✔ |
| Hamouda et al. [27] | ✗ | Small | Multi | ✗ | ✗ | ✔ |
| Bayrakdar et al. [28] | ✗ | Small | Multi | ✔ | ✗ | ✔ |
| Proposed work | ✗ | Large | Multi | ✔ | ✔ | ✔ |

There are several challenges that occur in wireless sensor networks; for example, complex routing algorithms share information with the neighboring nodes, which is a cause of congestion. Therefore, in order to increase the network's lifetime, traffic must be decreased [16]. In a large area, coverage is the main issue that occurs for wireless sensor networks [29]. The communication range also needs to be prolonged by adopting multi-tier ad-hoc techniques [30]. Similarly, fault tolerance is also needed in far-away deployment areas because several network failures occur due to physical damage, radio interference, blockage, and collision. The performance of a wireless sensor network also depends on

the sensor's size and placement. Random deployment should be used in large-scale areas, while the deterministic placement approach is better for small-scale deployment fields. [31]

## 3. Proposed System

An agent is composed of properties and characteristics based on which they interact with the environment; the characteristics are individual and independent based on the information from its surroundings.

Agent-based modeling (ABM) is a holistic approach to view an environment with the combination of agents with a set of interaction rules. Agents have five characteristics: perception of the environment including other agents, communication and interaction with the environment and other agents, memory of previous actions, knowledge of a policy comprising a set of rules, and heuristics or strategies.

Computer-assisted decision support systems are an emerging area to increase agricultural production with efficient resource usage [3]. Traditionally, such farm management systems are knowledge-based, using rules and guidelines to achieve goals and to define constraints for a particular farm or producer [32]. The proposed solution is implemented using an agent-based model in an AnyLogic environment. The IoT sensor nodes are deployed in multiple farms (roughly $370 \times 330$ m). The sensor nodes are static, they can be deployed at multiple locations. Sensor nodes have varying energies, and other than sensing the nodes, also act as a relay node to forward data towards the base station. The base node is deployed at the edge of the farm which is depicted below in the block diagram (Figure 1). The base node gathers all the data and provides a platform for prediction or analysis. The network connections between farms and base stations are through deployed sensor nodes. The deployed sensor node sends the data to the base station through the optimal path selected based on the residual energy. The residual energy is calculated for each node, and path decision is made on the basis of that residual energy. The main aim is to provide sustainability for data dissemination in a large-scale farm.

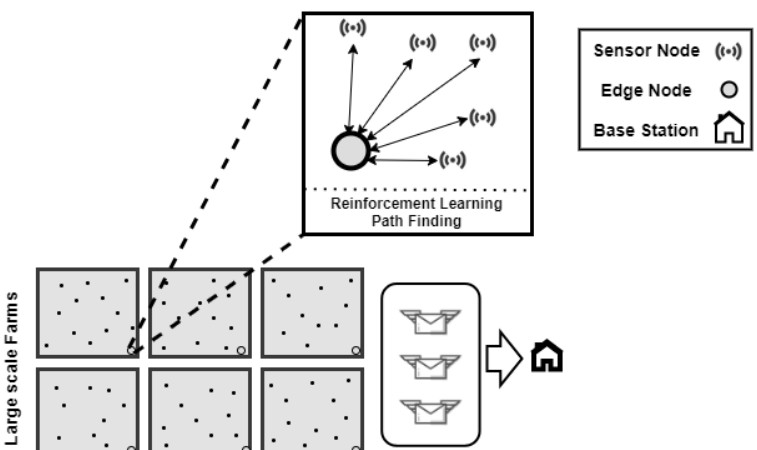

**Figure 1.** Large-scale farming scenario, base node is deployed at the west region.

### 3.1. Basic Elements

In the proposed work, we have used the reinforcement learning algorithm (Algorithm 1, defined below) to establish multiple data dissemination paths based on the residual energy of the deployed IoT sensor nodes. The residual energy information is used to select the path that shows more stability for data transmission from the sensory data towards the base station. The details of the main agents/components used in the proposed work are listed below; main agent is the entire simulation environment that includes all simulation operations. It is like the main class of any program that holds all core functionalities of the program. This agent has all the basic functions of simulation, and all other agents are either defined in this class or initialized in this class. We initialized farms, parameters and basic machine learning

state charts as well as code in the main agent. The message agent is designed to hold the messages that contain the sensor id, sensor time, and energy consumption, as well as the field parameters such as pH and temperature. The type of data a message contains is a floating point of number values. This agent is meant to take information from the sensors and direct it towards the base station by travelling hop by hop through the network. In order to perform the experiments in real life for future reference, the partition is a 370 × 330 m field/farm agent that is created for equal-sized fields which contain the sensor agent. Figure 1 represents the block layout of the model. This agent represents the field properties that held sensors deployed in it, including the edge node placed at one of its corners. Refer to Figure 1 for edge node where the number represents the farm number having sensors deployed.

The sensor is defined to simulate the behavior of sensor nodes. It resides inside the partitioning agent. The sensor agent resembles a real-life sensor because it has sensor cycles e.g., sleep, read, transmit, and dead states, including a battery that is consumed while performing its cyclic functions. The Algorithm 1 explains the algorithm used in the model while sensor cycles are explained in Algorithm 2 below. The sensors have contained a population of equal battery size and have their own specific Id. All sensors are initialized with same sensing capabilities and parameters such as pH and Temperature. The sensors are randomly placed within the connection range of each other. Since they are wireless, the connection range is up to 100 feet.

A pathfinder agent is responsible for path finding using a reinforcement learning algorithm. This agent checks the presence and availability of sensors, and connects them with the target node for successful attempts. This agent moves from sensor to sensor in search of path that can be connected with the edge node. If there is a success, the reinforcement algorithm gives an award and saves the path. In case of failure, punishment is given. Mathematical details are shared in Section 3.1.

A base station agent is defined to gather all the messages for prediction or analysis purposes. There is only one base station defined for the entire large-scale farming system. This agent has a database that stores all sensing data coming from the fields and exports it as an excel sheet to further analysis.

### 3.2. Reinforcement Learning Algorithm

There are four main concepts in reinforcement learning, namely the state (s), action (a), reward (R), and discount factor (γ). In Algorithm 1, the current situation of an agent is called the state (s). In our model the state (s) is the position of deployed nodes and pathfinding. Whatever an agent does is taken as an action (a). There are defined sets of actions that an agent can take. In any given state (s), the pathfinder can try to connect to the nodes lying in its proximity and try to find a path connected with the edge node. Upon taking any action (a) the agent receives an award (R), either positive or negative. For example, in our model, the pathfinder tries to find a path between edge node and deployed nodes. If there is a connected path found between several nodes, then the pathfinder is given green bars connecting those nodes as success. If there is no path or longer routes available among the nodes and the edge node, then that path receives red bars as a penalty. After several iterations there is a mesh of an optimal set of green paths and also a set of red paths. The discount factor (γ) tells us how much a feedback is given back to the subsequent movements of the learning. Because we will also be using the previously developed paths, we have set the discount factor up to a 0.8 value. The developed paths are saved in the Q table maintained by every node; therefore, later lookup can be performed. The Q table helps to find desired future rewards against every action an agent performs. The reward helps in choosing the best available option. The Q table learns values through iterative processes by using a Q-learning algorithm, which uses Bellman's equation:

$$V\left(s\right) = max_a(R(s, a) +_\gamma V\left(s'\right))\tag{1}$$

**Algorithm 1** Reinforcement Learning Algorithm

1: **procedure** REINFORCEMENT –LEARNING
2: **while** (reward = 0):
3:     a = chooseActionByEpsilon(s)
4:     track.add(<s,a>)
5:     s = s.takeAction(a)
6:     s = environment.act(s)
7:     reward = getReward(s)
8: **EndWhile**
9:/* update Qs with track and reward */
10: index = track.length
11: **end procedure**

After gradual iterations and rewards are achieved, the Q value is updated and the agent uses more of the developed policy for optimizing its policy in less time. For example, the value of any given state equals max action i.e.,

$$V\,(s) = max_a \tag{2}$$

- The action with maximizing value will be taken.
- The reward of optimal action a in any state s is taken and added with a multiplier of $\gamma$ which is a discount factor that reduces the reward by time.
- Each time an agent takes an action, it gets the next state to proceed i.e., $s'$
- As it is a recursive model, it takes $s'$ and puts it into V(s).

This is how the number of passes is counted and the Q table is updated each time an agent takes an action. The actions with maximum reward towards the target node are stored as the paths towards the destination (from each sensor individually). Hence, a mesh of optimal paths is developed as a result of our reinforcement algorithm training process. The training is conducted individually for each field. As soon as the pathfinding process is complete, we use these paths to send data from each sensor from the field to the base station.

**Algorithm 2** Sensor Duty Cycle Algorithm

1: **procedure** DUTY –CYCLE
2: **for** each x time where x − 10 s **do**
3:     Check the data
4:     Send the data
5:   **end for**
6:   **for** each y time where y − 3 min **do**
7:     sleep
8:    **end for**
9: **end procedure**

*3.3. Dynamics*

Algorithm 2 shows the sensor cycle adopted for the smart farming framework. The sensors have multiple stages such as read, sleep, and transmit. The sensor has a predefined energy model which is consumed during active mode. The transition between states is defined to gather the data at discrete points.

Battery dynamics: In AnyLogic, every single entity is taken as an agent, and for battery life there are agents or units; we set 1000 battery units for each sensor. The sensors were set to use 4 units for reinforcement algorithm execution. Sensors used 3 units for send state, 2 units for read state and 0.5 units for sleep state. The threshold level was set to 10 units and the sensor node was considered dead at less than 4 units left in the battery. The typical formula used for energy consumption is [33,34]:

$$x - (s_{use} + s_{transmission}) \tag{3}$$

where: x = battery units, $s_{use}$ = energy consumed for sensor cycles (send + read + sleep), $s_{transmission}$ = energy consumption during path finding and transmission of the data.

## 4. Evaluation

In this section, we describe the experimental setup and evaluation of the proposed framework. The proposed framework is implemented using AnyLogic. Table 2 lists the simulation parameters along with the input variables and system specifications used for the evaluation. The performance is evaluated in terms of delay, sensor battery, and amount of data transferred, by comparison with the traditional approaches (which are based on broadcast strategy—consuming high energy—and the shortest path first model). The experimental setup contains 6 fields with a range of 22 to 72 sensors each, randomly places, and located to the west of the base station. The benchmarking process is performed by increasing the sensor nodes to observe their impact on the proposed framework. In the proposed work, the farm's layout is shown in Figure 2, and the base station is placed at one end. The main objective is to disseminate the data to the base stations for more informed decision-making on the received data. The sensors have a limited battery; therefore, the communication or data relay reduces their life. Thus, the broadcast is not a viable approach in this scenario, as it can rapidly deplete the energy. The comparison is performed alongside the traditional approach which is based on broadcast strategy, the shortest-path first model, and the reinforcement learning-based work. In the evaluation, the same configuration setting is used for all three approaches.

**Table 2.** Simulation configuration and specification.

| Parameter | Value | Units |
|---|---|---|
| No. of sensors | Variable | sensors |
| Acquisition time | 10 | seconds |
| Transmit time | 5 | Seconds |
| Sleep time | 3 | minutes |
| Power to transmit | 1 | E/minute/meter |
| Number of fields | 6 | |
| Field size | 370 × 330 | m |
| Model time scale unit | - | minutes |
| **Set** | **Sensor/field** | **Sensors in 6 fields** |
| S1 | 22 | 132 |
| S2 | 32 | 192 |
| S3 | 42 | 252 |
| S4 | 52 | 312 |
| S5 | 62 | 372 |
| S6 | 72 | 432 |

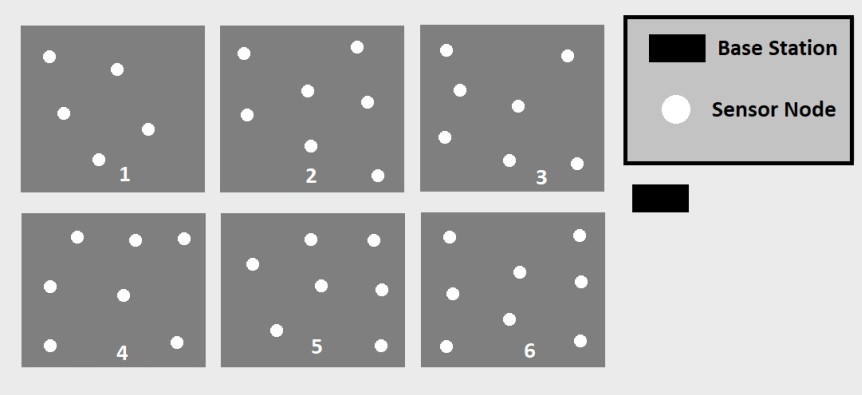

**Figure 2.** Geographic layout of the farm fields marked with numbers. The base station is located at the west side, and collects all the data for the decision.

The entire data flow from farms to base stations is relayed on the ground sensors; therefore, the system is evaluated with a various number of sensors. Thus, six sets have been created with a different number of sensors. In the first set, 22 sensors have been deployed on each farm. In every set, 10 additional sensors have been added, this is performed to find the acceptable network life for the data transfer in farming. Thus, set 6 contains a total of 72 sensors. For every set, a random deployment model is used.

### 4.1. Transmission Delay

The transmission delay is a crucial parameter used to gauge system performance. Figure 3 shows the transmission delay in different sets of sensors. The machine learning approach proposed in the paper shows a smaller transmission delay in all fields, lower energy consumption than broadcast, but worse for all except field 6 which has the most sensors (72). The machine learning approach also has the best message delivery rate. These experiments and results are indeed very relevant to the field. The communication path relies on the deployed sensors having limited energy. The transmission delay is computed for all the experiments performed with S1 to S6. Thus, the average value of the entire simulation run is reported here. In all these experiments, the proposed techniques have shown considerable improvement compared to the broadcast and shorted paths. Figure 3 below for visual representation for average transmission delay per message for different sensor sets.

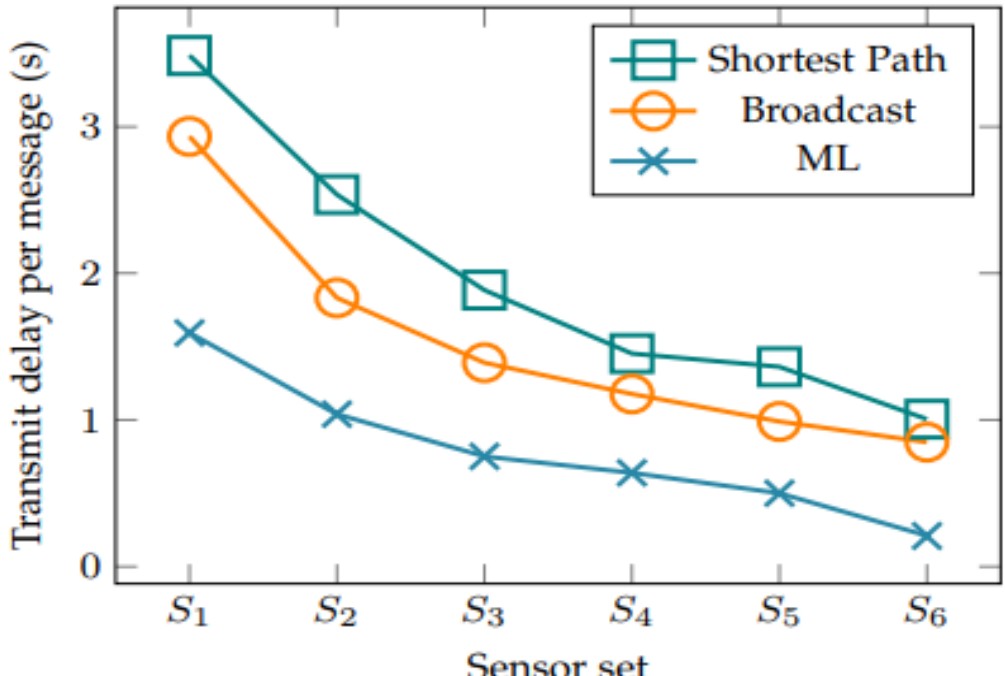

**Figure 3.** Average transmission delay per message for different sensor sets.

### 4.2. Energy Consumption

The energy consumption is an indicator to measure the lifespan of a network. It is directly related to the involvement of intermediate sensors; the frequent message exchange can deplete the energy and result in disjoint networks. Figure 4 for visual representation for average energy consumption for different sensor sets and while Figure 5 shows the energy consumption for all three techniques compared on S1 to S6.

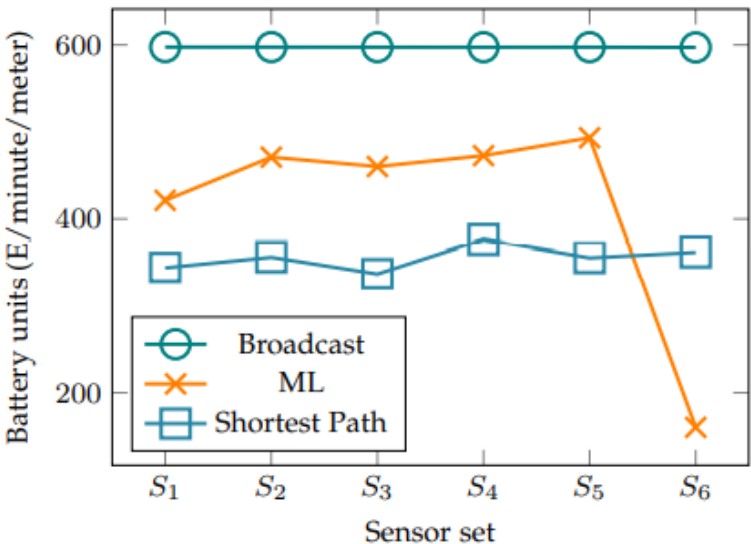

**Figure 4.** Average energy consumption for different sensor sets.

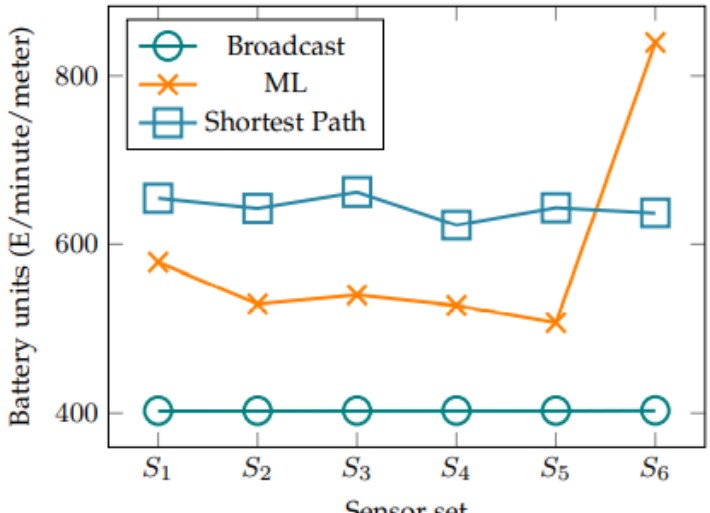

**Figure 5.** Average residual energy after sending messages for different sensor sets.

Thus, an increasing number of sensors provide an alternative route for data transfer that results in reduced energy consumption. The shortest path approach utilizes the same path, and thus saves energy by avoiding frequent path searching. On the other hand, in broadcast, all sensors are involved, thus having maximum energy consumption. The proposed work relies on the reinforcement model which selects the path based on the given features; therefore, it lies between the two extremes.

### 4.3. Average Residual Energy

The plot below shows the residual energy for different sensor sets after they sent a specific number of messages. This calculation was performed on a battery having 1000 units; for each field average, residual energy was taken. It can be seen that no matter how much we increase the field scale and number of sensors, there is not much energy conservation seen in broadcast and shortest path transmission techniques. However, in our proposed transmission technique, it conserves energy as we increase the number of sensors over a large scale. This trend, however, was seen only after sensor set 5 and sensor set 6.

### 4.4. Packets Delivered

This parameter is used to see the network reliability. Figure 6 shows the comparison of all three techniques in terms of delivery rate. The proposed work shows a better delivery rate among other approaches due to the optimal path selection based on the residual energy. Further, it constructs many different paths to the base stations, and switches to them based on the optimal use of these resources. However, the other two approaches show a lesser delivery rate compared to the base station. Figure 6 shows packet delivery rate for S1 to S6 deployment.

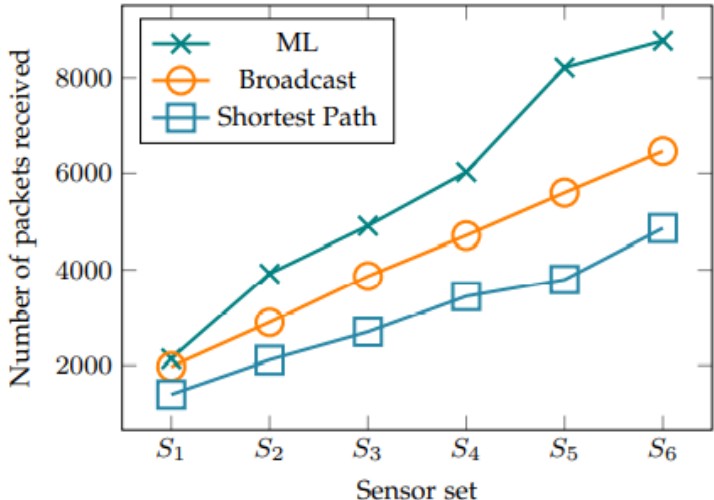

**Figure 6.** Packet delivery rate for S1 to S6 deployment.

### 5. Conclusions

Our main goal pursued in this research work was to develop an efficient transmission system in terms of energy and time. Efficient energy consumption is an emerging area of research. In previous work, researchers have mostly focused on sensor technology rather than the scaling and performance of the sensors when deployed on a large scale. Some of the work was conducted to test performance, but on small-scale networks including less than 20 sensors in the network, our work mostly focuses on transmission delays and energy-hole problems. The performance results of the machine learning-based method on a large scale are better than the traditional broadcasting and shortest-path message-sending techniques. The proposed system shows the energy-efficient transmission of sensory data in large-scale farming by using paths created through reinforcement learning. The proposed model is based on reinforcement learning that uses random deployment topology for sensors as an input parameter, and creates several optimal paths for data transmission, fundamentally intended to reduce the energy consumption in a wireless sensor network meant for precision agriculture on a large scale. The objectives of the proposed transmission model are energy consumption minimization, reduction in average transmission delay, and ultimately to reduce the management process for large-scale wireless sensor networks, especially by increasing the network lifetime.

Some of the gaps in this research are pointed out as future work, and include the fact that to implementing this system in large-scale farming networks accompanied by artificial intelligence would be helpful in increasing efficiency. Implementing and experimenting with more topologies to test energy efficiency is also necessary. The proposed model also has room for optimization in terms of how data is being processed, including the sensor cycles. Moreover, it might be possible to make this model adaptive in precision agriculture after added work is conducted. The model can also be extended with minor modifications to ad-hoc networks and vehicular networks.

**Author Contributions:** Conceptualization, M.Y.A., W.M.S.Y. and A.A.; methodology, all authors; software, W.M.S.Y., M.Y.A. and S.A.A.S.; validation, all authors; formal analysis, S.A.A.S., A.W.M., A.A., W.M.S.Y. and S.A.A.S.; investigation, W.M.S.Y., A.W.M. and A.A.; resources, all authors; data curation, all authors.; writing—original draft preparation, writing— review and editing, all authors; visualization, W.M.S.Y. and A.A.; supervision, S.A.A.S., W.M.S.Y. and A.W.M. All authors have read and agreed to the published version of the manuscript.

**Funding:** This research received no external funding.

**Acknowledgments:** The authors would like to thank National University of Science and Technology (NUST) and AIR University for providing resources for the project.

**Conflicts of Interest:** The authors declare no conflict of interest.

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
