# Peer review of "Energy Efficient Data Dissemination for Large-Scale Smart Farming Using Reinforcement Learning"

_electronics, doi:10.3390/electronics12051248_

Round 1
Reviewer 1 Report
The authors have done their very best and I commend their efforts and innovation.
My comments are as follows:
Lines 91 should be moved to the next page for proper structuring of the paper.
Lines 169 should be moved to the next page for proper structuring of the paper.
Lines 259, should be moved to the next page for proper structuring of the paper.
Figure 2 can be captured better and clearer.
The references are ok but I suggest the author look at broader literature that dealt with similar areas (AI/ML).
A Machine Learning Method with Filter-Based Feature Selection for Improved Detection of Chronic Kidney Disease” Bioengineering 2022, vol. 9, no. 8, 350; https://doi.org/10.3390/bioengineering9080350, Switzerland.
Integrating Enhanced Sparse Autoencoder Based Artificial Neural Network Technique and SoftMax Regression for Medical Diagnosis Published MDPI Electronics Journal,2020, 9(11), 1963; https://doi.org/10.3390/electronics9111963 Switzerland.).
Author Response
Dear Reviewer,
Thanking you for your consideration and for providing us with a detailed review. All the commented issues are fixed and are responded to below.
Comment 1: Line 91 should be moved to the next page for proper structuring of the paper..
Response: This observation is correct, the lines have been readjusted.
Comment 2: Line 169 should be moved to the next page for proper structuring of the paper.
Response: the lines have been readjusted.
Comment 3: Line 259, should be moved to the next page for proper structuring of the paper.
Response: We thank the reviewer for pointing this out. It is readjusted.
Comment 4: Figure 2 can be captured better and clearer.
Response: We replaced it with fine quality image.
Comment 5: The references are ok but I suggest the author look at broader literature that dealt with similar areas (AI/ML).
Response: Thank you for your valuable suggestion. Such changes will broaden the scope of the paper and would be difficult to accommodate in this paper. However, We are trying to improve our knowledge with more detailed AI/ML techniques which can be applied to observe behaviour of our model. The details will be adjusted in the latest AI/ML agricultural model we are working with in another paper. .
Comment 6: Comments and Suggestions for Authors
Response: Moreover, there were several comments that have been given to the authors to adjust the paperwork which has been added to the text.
We are warmly thankful to the reviewers for the valuable suggestions and appreciate their in-depth analysis that helped us to improve the quality of this paper.

Reviewer 2 Report
The paper is concerned with improving wireless sensor communication in large farms that utilised IoT technology to monitor the crop conditions. This is performed with the help of machine learning algorithms, more specifically Reinforcement Learning, applied on a large network of sensors, with the purpose to improve communication with the control station. The proposed solution manages to find successful path to the base station with a lesser delay than the shortest path transmission model.
Some of the literature review focus on low power wireless sensor networks, due to the fact that data transmission is often energy consuming, and sensors reply on batteries or researchable power which is limited. This is performed by using specific node-based transmission or sampling rates, with different algorithmic approaches. Other studies focus on network configuration or lifetime optimisation to avoid network congestion.
The proposed study aims a sustainable wireless sensor network from all perspectives (energy, communication, configuration), by training agents through reinforcement learning to find an optimal path, based on specific rewards (like the shortest path from sensor node to edge node), taking in consideration energy spend when transmitting data (also ‘rewarding’ agents with units).
The performance is evaluated in terms of delay, sensor battery, and amount of data transferred, by comparison with the traditional approaches (which are based on broadcast strategy – consuming high energy, and the shortest-path first model). The experimental set-up contains 6 fields with a range of 22 to 72 sensors each, randomly places, located to the west of the base station.
The ML approach proposed in the paper shows smaller transmission delay in all fields, lower energy consumption than broadcast but worse for all except field 6 which has the most sensors (72). ML approach also has the best message delivery rate. These experiments and results are indeed very relevant to the field.
Some details could greatly improve and highlight the importance of this research. The size of the farm fields set-up is not specified and I believe this is relevant for the real-life experiment. Perhaps you can provide more details on the types of sensors used, the placing of the sensors (even if random, what are the distances) and the amount and type of data transmitted and over what period of time. Moreover, it would be very useful to see how these characteristics and results compare with state-of-the-art, if this is possible. Several related works are cited in section 2 – could any of those experiments be compared to these in any way? A discussion on the results and highlighting their importance and novelty in relation to related work would be helpful.
The paper also requires refinement of the English language used, in terms of verbs, transition words, and prepositions, perhaps some rephrasing to improve readability and understanding of the content.
The references are relevant, most of them are recent and represent scientific work related to the field.

Author Response
Dear Reviewer,
Thanking you for your consideration and for providing us with a detailed review. All the commented issues are fixed and are responded to below.
1) The size of the farm fields set-up is not specified and I believe this is relevant for the real-life experiment.
Response: Formatted accordingly, the farm size might appear unclear in the discussion but we have defined 370 x 330 meters field/farm agent in the model which is defined in section 3.1 “Basic Elements.”
2) Perhaps you can provide more details on the types of sensors used,
Response: Thank you for the valuable suggestion. The sensor details was missing from the model and it is added after this comment. The sensors are temperature and di-electric soil moisture sensors.
3)The placing of the sensors (even if random, what are the distances)
Response: The sensors are placed within the connection range of each other. Since they are wireless the connection range is up to 100 feet. It was not mentioned in the paper but after the comment, it is now added in the paper.
4) The amount and type of data transmitted and over what period of time
Response: Yes, the type of data a message contains is a floating point number value. Also added in the definition of message agent.
5)Moreover, it would be very useful to see how these characteristics and results compare with state-of-the-art, if this is possible.
Response: Yes, we already have compared our proposed model with the broadcasting model and shortest path model. Other models that are discussed in the literature review are under consideration. It requires another study to study their behaviours as compared to our model. It will be compared in another version of this work with some other changes in the placement of sensors as well.
6) Moreover, it would be very useful to see how these characteristics and results compare with state-of-the-art, if this is possible.
Response: Yes, as we discussed in the previous comment, the related work’s experiments can be compared and need to be modelled in AnyLogic which is a whole other research work to model and compare. We are working to model their behaviours as well.
7)A discussion on the results and highlighting their importance and novelty in relation to related work would be helpful.
Response: In previous work, researchers have mostly focused on the sensor technology rather than the scaling and performance of the sensors when deployed on the large scale. Some of the work was done to test the performance but on the small-scale networks including less than 20 sensors in a network, our work mostly focuses on transmission delays and energy-hole problems. The performance results of the Machine Learning based method on large scale are quite better than the traditional broadcasting and shortest-path message-sending techniques. This text is added under the conclusions heading.
8). The paper also requires refinement of the English language used, in terms of verbs, transition words, and prepositions, perhaps some rephrasing to improve readability and understanding of the content.
Response: The paper is revised and changes have been made, we found some grammatical mistakes. Thank you for reviewing this paper carefully.
We are cordially thankful to the reviewers for the valuable suggestions and appreciate their in-depth analysis that helped us to improve the quality of this paper.
